# The energy-efficient reductive tricarboxylic acid cycle drives carbon uptake and transfer to higher trophic levels within the Kueishantao shallow-water hydrothermal system

Joely M. Maak[1], Yu-Shih Lin[2], Enno Schefuß[1], Rebecca F. Aepfler[1], Li-Lian Liu[2], Marcus Elvert[1,3], and Solveig I. Bühring[1]

[1]MARUM- Center for Marine Environmental Sciences, University of Bremen, Bremen, 28359, Germany
[2]Department of Oceanography, National Sun Yat-sen University, Kaohsiung, 80424, Taiwan
[3]Faculty of Geosciences, University of Bremen, Bremen, 28359, Germany

*Correspondence to*: Joely M. Maak (jmaak@marum.de), Solveig I. Bühring (sbuehring@marum.de)

**Abstract.** Chemoautotrophic Campylobacteria utilize the reductive tricarboxylic acid (rTCA) cycle for carbon uptake, a metabolic pathway that is more energy efficient and discriminates less against $^{13}$C than the Calvin-Benson-Bassham cycle. Similar to other hydrothermal systems worldwide, Campylobacteria dominate the microbial community of the shallow-water hydrothermal system off Kueishantao (Taiwan). Compound-specific carbon stable isotope analyses of lipid-derived fatty acids were performed to understand the importance of rTCA and the transfer of fixed carbon to higher trophic levels in the vent area. Of these, $C_{16:1\omega7c}$, $C_{18:1\omega7c}$, and $C_{18:1\omega9c}$ fatty acids were strongly enriched in $^{13}$C, indicating the activity of rTCA utilizing Campylobacteria. Isotopic fractionation was close to 0‰, likely caused by pH values as low as 2.88. Characteristic fatty acids were present not only in the vent fluids but also in adjacent sediments and water filters 20 m away from the vent orifice, even though with decreasing abundance and diluted $^{13}$C signal. Furthermore, $\delta^{13}$C analysis of fatty acids from the tissue of *Xenograpsus testudinatus*, a crab endemic to this particular vent system, identified the trophic transfer of chemosynthetically fixed carbon. This highlights the interrelationship between chemoautotrophic microbial activity and life opportunities of higher organisms under environmentally harsh conditions at shallow-water hydrothermal systems.

## 1 Introduction

Shallow-water hydrothermal systems are found worldwide and can strongly impact the photic zone of coastal environments due to their emanation of often acidic, hot, and $CO_2$-rich fluids (e.g., Price and Giovannelli, 2017; Caramanna et al., 2021). Microorganisms with vast metabolic flexibility can adapt to these extreme environments (e.g., Tarasov et al., 2005; Sollich et al., 2017; Zeng et al., 2021). They can profit from the reduced chemical compounds and emitted carbon through chemoautotrophy, a process described as the most important carbon fixation pathway in many shallow-water hydrothermal systems (Gomez-Saez et al., 2017; Sievert et al., 2022).

The volcanic islet Kueishantao, located at the southern end of the Okinawa Trough (Lee et al., 1980), shows strong venting activity, with about 30 vents emitting hydrothermal fluids in water depths of 10 to 30 m (Chen et al., 2005b). The emitted

gases are dominated by $CO_2$ (~92%) with up to 8% $H_2S$ (Yang et al., 2005). This shallow-water hydrothermal system shows distinct environmental properties with highly acidic pH values of down to 1.5, high temperatures of up to 116°C, and high sulfur emission rates (Chen et al., 2005a; 2005b). Generally, two different subsystems are present in this area: active yellow vents (YV, also called high-temperature vents) and semi-inactive white vents (WV, also called low-temperature vents), with

the YV showing more acidic pH values, higher temperatures, and higher fluid emission rates (Chen et al., 2005a; 2005b). Faunal diversity changes according to the toxicity of the environment. Directly at the vent systems, the highly adapted crab *Xenograpsus testudinatus* (Ng et al., 2000, Allen et al., 2020) prevails, while at a few tens of meters distance, generalist species can occur as well (Chan et al., 2016; Chang et al., 2018).

The microbial community within the plume of shallow-water hydrothermal systems is characterized by bacteria, with the

dominant groups being Campylobacteria (formerly Epsilonproteobacteria; Waite et al., 2017) and Gammaproteobacteria (Wang et al., 2015; Li et al., 2018). Within Campylobacteria and Gammaproteobacteria, the most abundant members include chemoautotrophic *Sulfurimonas*, *Sulfurovum*, *Thiomicrospira*, and *Thiomicrorhabdus* (sulfur-oxidizers) and *Nautilia* (sulfur-reducers; Zhang et al., 2012; Wang et al., 2015; Li et al., 2018). Metagenomic investigations at Kueishantao show a high abundance of key enzymes for the reductive tricarboxylic acid (rTCA) cycle for $CO_2$ fixation in the vent fluids compared to

the surface waters above (Tang et al., 2013). The rTCA is a more energy-efficient alternative to the Calvin-Benson-Bassham (CBB) cycle, a clear advantage in the potentially energy-limited environments of shallow-water hydrothermal vent systems (e.g., Campbell and Cary, 2004). Since the rTCA cycle discriminates much less against $^{13}$C (isotope fractionation factor ($\varepsilon$) is 2 to 13‰ for rTCA and 10 to 22‰ for CBB; Hayes, 2001; House et al., 2003), the biomass and activity of microorganisms performing this cycle can be traced by their enriched stable carbon isotope compositions (e.g., Chang et al., 2018).

Campylobacteria are generally known to use the rTCA for chemoautotrophic carbon fixation (Hügler and Sievert, 2011; Waite et al., 2017), although one symbiotic species was recently shown to use the CBB cycle (Assié et al., 2020).

The fatty acid distribution and their stable carbon isotope composition ($\delta^{13}$C values) have been investigated in sediments of other shallow-water hydrothermal systems (e.g., Callac et al., 2017) to quantify the contribution of rTCA cycle-mediating microbes. Still, to date, no such studies have been conducted on the shallow-water hydrothermal systems off Kueishantao. In

the present study, we address how extreme conditions with strongly acidic pH values and high gas and fluid emission rates (e.g., Chen et al., 2005a; 2005b) imprint on the carbon fixation pathway and the activity of Campylobacteria. Furthermore, we track the carbon transfer from rTCA-utilizing chemoautotrophs into higher trophic levels by investigating tissues from the endemic vent crab *Xenograpsus testudinatus*. We performed a detailed organic geochemical characterization of vent fluids, hydrothermal sediments, *X. testudinatus*, and filtered particulate organic carbon (POC) from the water column (Fig. A1, Tab.

A1, Mw series at WV and My series at YV), specifically aiming at the distribution of fatty acids and their corresponding $\delta^{13}$C values.

## 2 Results

The fluids of the YV and WV sampled for this study show temperatures of 116 and 58°C and pH values of 2.88 and 4.51, respectively (Lin et al., 2019). Generally, background water-derived dissolved inorganic carbon (DIC) displayed $\delta^{13}C$ values of -0.35‰, while the DIC analysis from the fluids at the YV and WV sites resulted in values of -6.7 and -6‰ (Lin et al., 2020). Bulk carbon isotope analysis of tissue material of *X. testudinatus* resulted in $\delta^{13}C$ values of -19.3‰ (muscle material) and -20.7±0.04‰ (stomach material).

Fatty acids mostly comprise chain lengths from $C_{12}$ to $C_{18}$ with $C_{20}$ and $C_{22}$ present in some samples (Fig. 1). In the sediments, most of the filtered POC samples, and the white vent fluid, $n$-$C_{16:0}$ shows the highest abundance with 120 to 2300 ng g$^{-1}$ in the sediments and 770 to 5500 ng L$^{-1}$ in the vent fluids and filtered POC. Exceptions are the YV fluid, the muscle and stomach material of *X. testudinatus* ($C_{18:1\omega7c}$, 7300 ng L$^{-1}$ and 570 x 10$^3$ ng g$^{-1}$ and 2200 x 10$^3$, respectively), and the background sediment M4 (10Me-$C_{16}$, 2300 ng g$^{-1}$). Terminally-branched and mid-chain branched fatty acids are present in most samples and comprise 36 to 1000 ng g$^{-1}$ (9.6 to 16% of all fatty acids, respectively) in the sediment samples and 17 to 310 ng L$^{-1}$ (1.9 to 16%, respectively) in the filtered POC. Terminally branched fatty acids are absent in the vent fluids and the tissue material of *X. testudinatus*. Contents of mono- and di-unsaturated fatty acids range from 50 to 2600 ng g$^{-1}$ (11 to 40% of all fatty acids, respectively) in sediments, and from 120 to 1900 ng L$^{-1}$ (17 to 41% of all fatty acids, respectively) in the filtered POC. In the vent fluids, monounsaturated fatty acids comprise 2800 and 9300 ng L$^{-1}$ (20 and 50%, respectively), whereas in the muscle and stomach material of *X. testudinatus*, monounsaturated fatty acids comprise 1300 x 10$^3$ and 5200 x 10$^3$ ng g$^{-1}$ (53 and 58%, respectively). The background samples (sediment M4 and POC filter B0) contain 2900 ng g$^{-1}$ (34%, M4) and 160 ng L$^{-1}$ (17%, B0, Fig. 1) monounsaturated fatty acids. Within the background samples, terminally-branched fatty acids are only present in the sediment (M4) with 3900 ng g$^{-1}$ (44%).

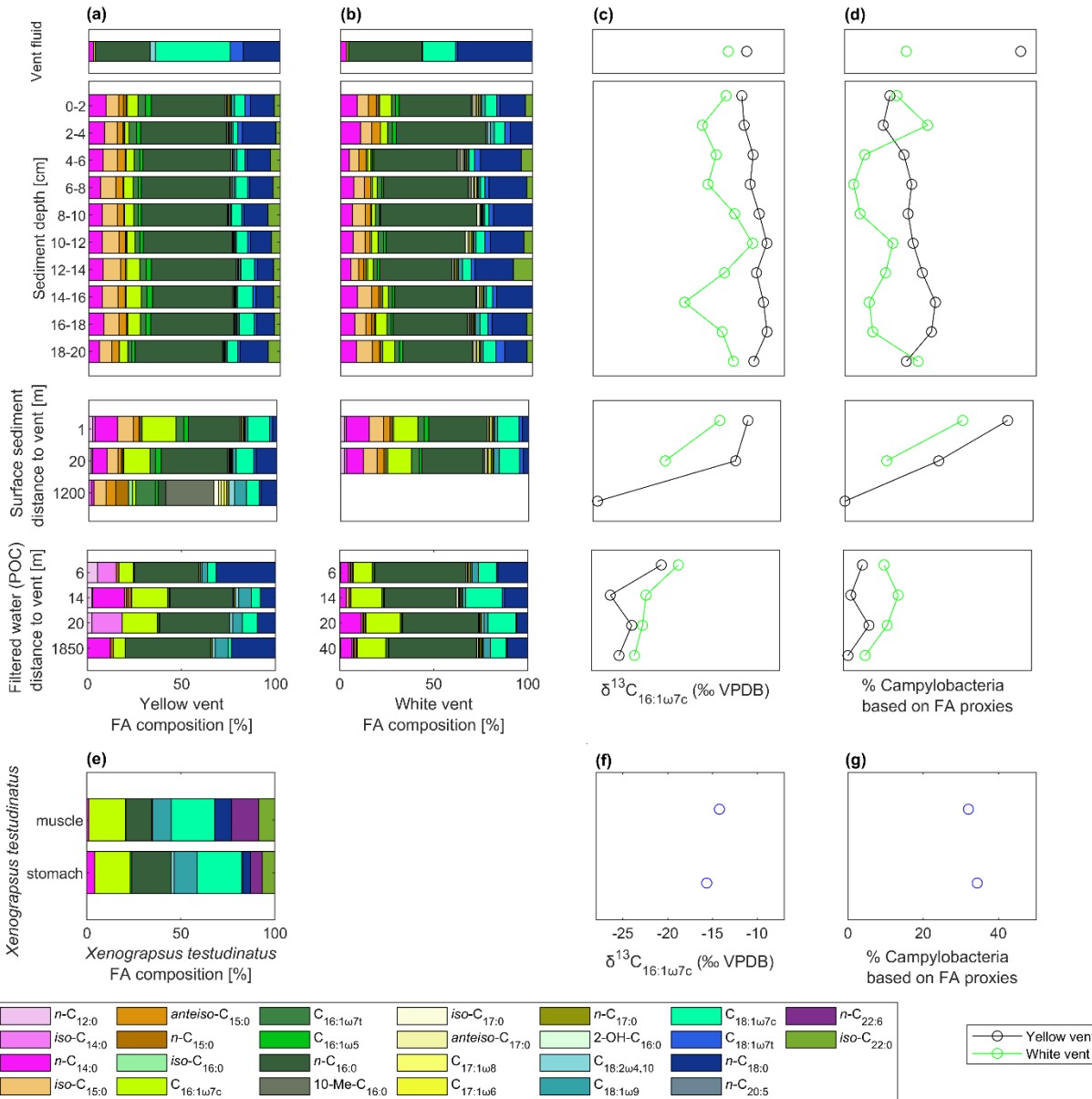

**Figure 1. Dominance of rTCA utilizing microorganisms closely to the vent orifice revealed by fatty acid (FA) composition of the yellow vent (a), white vent (b), and *Xenograpsus testudinatus* (e), including δ$^{13}$C values of C$_{16:1\omega7c}$ (c: yellow vent and white vent samples, f: *Xenograpsus testudinatus*), and % Campylobacteria in fatty acids based on the $^{13}$C values of their respective fatty acids (d: yellow vent and white vent samples, g: *Xenograpsus testudinatus*). Samples include the vent fluids, sediment cores, surface sediments in distance to the individual venting sites (including background site M4 in ca. 1200 m distance), filtered POC (including My series from the YV, Mw series from the white vent, and the background site B0 in ca. 1850 m distance, see Tab. A1), and stomach and muscle material of *X. testudinatus*. C$_{16:1\omega7c}$ and C$_{18:1\omega7c}$ are highlighted in brighter colors since they are most enriched in $^{13}$C.**

Compound-specific stable carbon isotope measurements revealed uniform $\delta^{13}C$ values for most fatty acids (mostly between -30 to -20‰) but a few fatty acids were identified that deviated from this range with more positive $\delta^{13}C$ values of -17.9 to -7.2‰ (Fig. 2). These included $C_{16:1\omega7c}$, $C_{18:1\omega9c}$, and $C_{18:1\omega7c}$ fatty acids found in samples taken closer to the vent (Figs. 1 and 2). The potential percentage of Campylobacteria, calculated as a mass balance from the $\delta^{13}C$ values of fatty acids produced by sulfur oxidizers present in the samples, gave the highest values in the vent fluid and surface sediments at 1 m distance from

YV (46% and 43%, respectively, Fig. 1). The lowest percentage of Campylobacteria (~2%, Fig. 1) was found in one section within the WV sediment core (6-8 cm) and the filtered POC at 14 m distance to the YV (My 14-5). The endemic vent crab *X. testudinatus* displays fatty acid $\delta^{13}C$ values ranging from -22.6 to -14.2‰ (Fig. 2). Using the mass balance approach, the percentage of fatty acids presumably originating from Campylobacteria comprises 34% in the stomach and 32% in the muscle material (Fig. 1). Within the background samples (M4 and B0), taken in a non-hydrothermally influenced area, no $\delta^{13}C$ values

above -20‰ are present (Fig. 2). The mean isotope signatures of all fatty acids are -27.9 ± 4.1‰ for the sediment (M4) and -25.6 ± 1.0‰ for the filtered POC (B0). To distinguish the degrees of hydrothermal influence in the different samples, a cluster analysis using the $\delta^{13}C$ values of diagnostic fatty acids present in all samples was used (Fig. 3). This revealed that the samples can be subdivided into three statistically significant groups (Fig. 3).

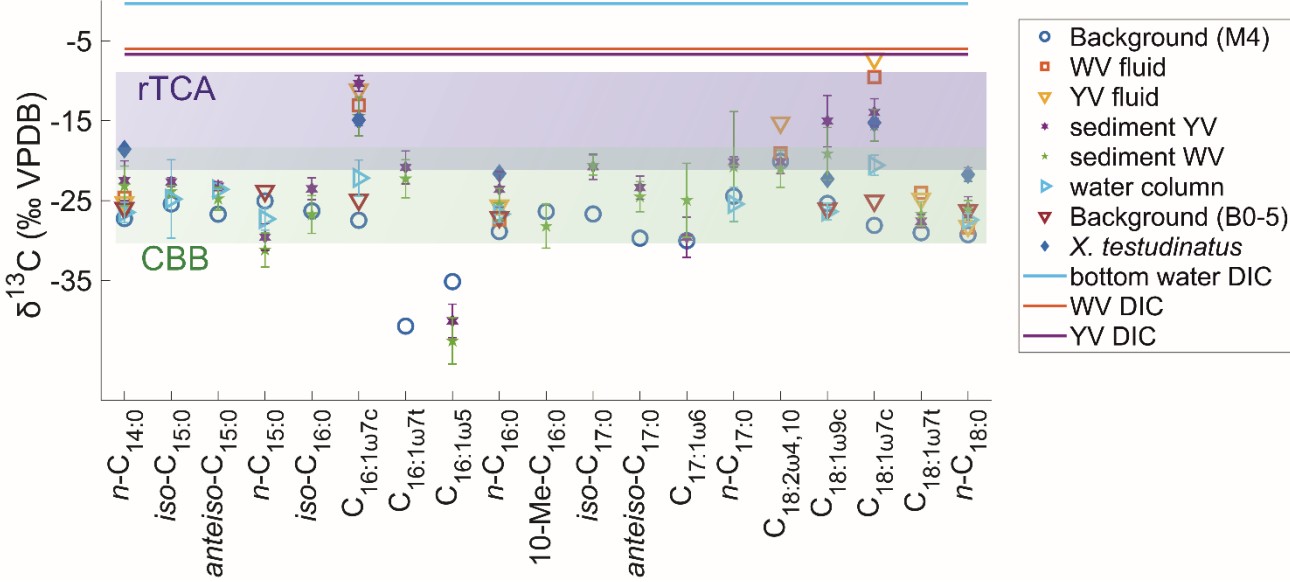

**Figure 2. Influence of rTCA on individual fatty acids displayed by their $\delta^{13}C$ values in surface and deeper sediments, vent fluids, tissue material of *Xenograpsus testudinatus* (stomach and muscle), and all filtered POC from the water column (taken 40 meters or less in distance from the venting sites, Tab. A1) from the white vent (WV) and the yellow vent (YV) system compared to background samples. DIC $\delta^{13}C$ values are from Lin et al. (2020); isotope fractionation (2 to 13‰ for rTCA – lilac, 10 to 22‰ for CBB – light green) are from Hayes (2001) and House et al. (2003) and express expected ranges of the resulting biomass. Since lipids can exhibit**
**greater isotope fractionation than other biomolecules such as proteins and carbohydrates (DeNiro and Epstein, 1977), a larger isotope fractionation than the indicated ranges is possible.**

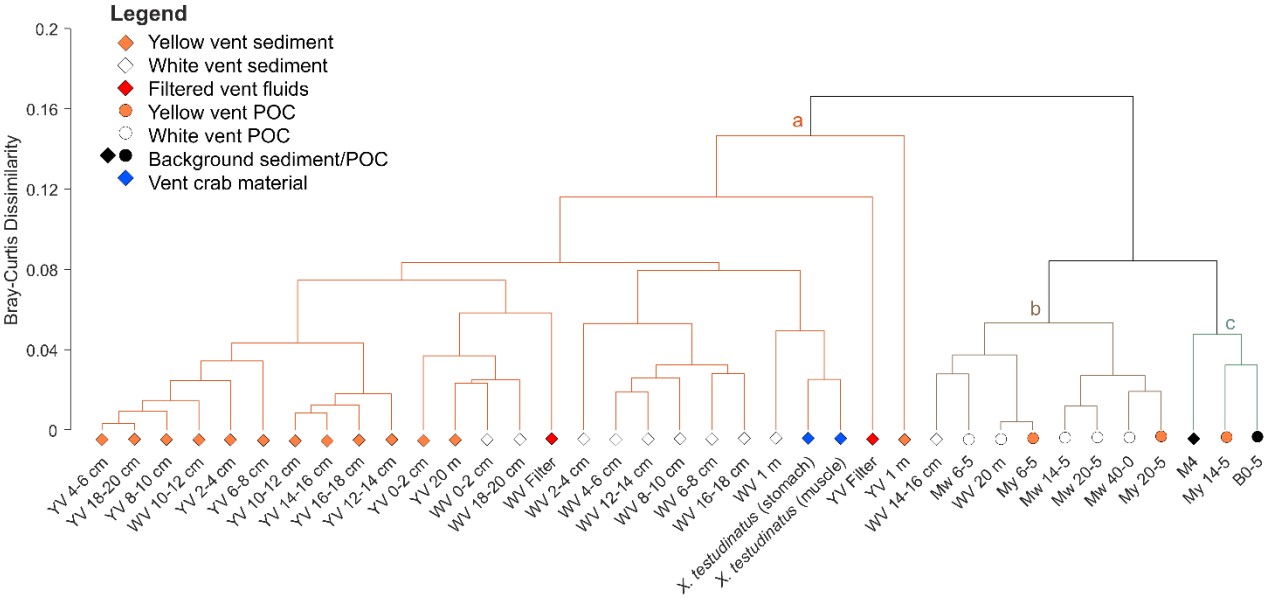

**Figure 3. Allocation of all samples analyzed in this study into clusters with decreasing hydrothermal influence using the $\delta^{13}C$ values of fatty acids typically synthesized by sulfur-oxidizers ($C_{16:1\omega7c}$, $C_{16:0}$, $C_{18:1\omega7c}$). The dendrogram together with the SIMPROF test identified three clusters that are statistically different (a = strong hydrothermally influenced sediments, vent fluids, and muscle and stomach material of *Xenograpsus testudinatus*, b = less hydrothermally influenced sediments and filtered POC, c = least/not hydrothermally influenced sediment and filtered POC). My = series of filtered POC samples taken close to the yellow vent (YV), Mw = series of filtered POC samples taken close to the white vent (WV).**

## 3 Interpretation and Discussion

Despite the extremely hot and acidic conditions, microorganisms thrive at the hydrothermal vent sites of Kueishantao. The abundance of rTCA enzymes has already been proven to be present in the associated hydrothermally influenced sediments (Tang et al., 2013; Wang et al., 2017). Our study now shows that certain fatty acids (namely $C_{16:1\omega7c}$, $C_{18:1\omega7c}$, and $C_{18:1\omega9c}$) in sediments, vent fluids, filtered POC, and in tissue material of *X. testudinatus* originate to a large part from rTCA utilizing bacteria (Figs. 2 and 3). Such fatty acids have been described in many cases as originating from Campylobacteria (e.g., Inagaki et al., 2004; Takai et al., 2006; Mino et al., 2014; Giovannelli et al., 2016), which possess the rTCA as central carbon metabolic pathway (Wang et al., 2021). When using rTCA, the isotope fractionation between the substrate and cell products, expressed as ε, is in the range of 2 to 13‰ (Preuß et al., 1989; Hayes, 2001; House et al., 2003). In theory, lipids can exhibit greater isotope fractionation than proteins and carbohydrates due to further biosynthetic processes (DeNiro and Epstein, 1977). Despite this potential variability, the indicated ranges provide a valuable framework for estimating whether samples or specific fatty acids can be assigned to particular carbon fixation pathways. With the individual DIC $\delta^{13}C$ values at the WV and YV venting sites of -6.7 and -6.0‰ (Lin et al., 2020) and $C_{18:1\omega7c}$ as most $^{13}C$-enriched fatty acid in the vent fluids, we obtain very low ε values of 3.6 and 0.5‰, respectively (Fig. 2). This is in accordance with the abundance of Campylobacteria, which are predominating in the YV fluid and are less abundant in the WV fluid (Tang et al., 2018; Chen et al., 2022). DIC concentrations

at the hydrothermal vents are in excess (Lin et al., 2020), despite the acidic pH values that typically promote $CO_2$ degassing.

The elevated concentrations are likely sustained by a constant supply of carbon-rich fluids from below. Therefore, it is unlikely that such a low isotope fractionation is caused by carbon limitation as suggested in previous studies from other hydrothermal sites (e.g., Bradley et al., 2009). Isotope fractionation below 2‰ has only been reported from fatty acids in cultured representatives once (Elling et al., 2022) and, so far, never from environmental samples. We argue, that environmental stressors such as pH or temperature could influence ε during rTCA, finally resulting in extremely low isotope fractionation. While

temperature effects on rTCA utilizing microbes have not yet been examined, *Thermoproteus neutrophilus* showed a strong decrease in ε in two growth experiments at different pH values (ε of 8.6‰ with a pH of 6.8 to 6.9 in Preuß et al., 1989, and ε of 2.0‰ with a pH of 6 in House et al., 2003). This would be in line with the most $^{13}$C-enriched signals of $C_{18:1\omega7c}$ in the YV fluid (-7.2‰), where the lowest pH was observed (2.88), while at the more moderate WV with a pH of 4.51, the same fatty acid is less $^{13}$C-enriched (-9.5‰).

Aside from the vent fluids, fatty acids from the sediment cores at the venting sites and surface sediments further away fall within the ε range for rTCA (Fig. 2). This is further supported by the results of a cluster analysis using fatty acid $\delta^{13}$C values as input parameters (Fig. 3), which underlines the different degrees of hydrothermal influence and rTCA activity within the samples. The first cluster includes all samples with the strongest visible hydrothermal influence and therefore lowest pH (vent fluids, sediment cores directly at the vent, surface sediments, tissue material of *X. testudinatus*). The second group includes

most POC samples and two sediments (WV 20 m and WV 14-16 cm), which are seemingly less hydrothermally influenced. The third group includes both background samples and one POC sample (My 14-5), where all samples show the least to no hydrothermal influence. Generally, our cluster analysis proves that the influence of the vent plume on the carbon uptake mechanism seems to cover a distance of at least 20 m from the discharge area (Fig. 3). Samples in the first cluster are mostly those at and near the YV site showing a stronger $^{13}$C-enrichment in fatty acids (Figs. 1 and 3), likely caused by the lower pH

values and the higher fluid emission rates compared to WV (e.g., Chen et al., 2016).

Campylobacteria were found to be highly abundant (≥50% of all major bacterial taxa) in the stomach, gill, mid-gut, and digestive gland of *X. testudinatus* (Yang et al., 2016). A symbiotic relationship is inferred from the presence of Campylobacteria in the gill tissues of *X. testudinatus* where they function as detoxifiers of hydrogen sulfide (Chou et al., 2023). Fatty acids in *X. testudinatus* have been examined before (Hu et al., 2012) and, similar to this study, showed $C_{16:0}$, $C_{16:1\omega7c}$,

$C_{18:1\omega7c}$, and $C_{18:1\omega9c}$ at highest abundance. Based on the positive fatty acid $\delta^{13}$C values of *X. testudinatus* of up to -14.2‰ (Figs. 1 and 2) we could show that Campylobacteria and their biomass components are an important diet for the crab. This finding is further supported by the tissue material of *X. testudinatus* being classified into the cluster of samples with the strongest hydrothermal influence and thus proportion of rTCA (Fig. 3). Additionally, bulk $\delta^{13}$C values of muscle (-19.3‰) and stomach (-20.7‰) material measured in this study are more negative than most of the $^{13}$C values previously reported, which range

from -20‰ to -13.9‰ (Chang et al., 2018, Hung et al., 2019, Wu et al., 2021, Wang et al., 2022, Wu et al., 2023). This suggests that previously analyzed specimens likely show a higher proportion of campylobacterial-derived biomass in the diet than indicated by bulk isotope approaches alone.

Through the process of assimilation, $\delta^{13}C$ values are expected to increase by 0 to 1.3‰ in each trophic transfer (e.g., DeNiro and Epstein, 1978; McCutchan et al., 2003). However, only those fatty acids that are associated with the sulfur-oxidizing Campylobacteria show more positive $\delta^{13}C$ values in the tissue of *X. testudinatus*, while other fatty acids provide $\delta^{13}C$ values of approximately -20‰ that resemble carbon fixed by CBB (Fig. 2). Further, a high lipid content originating from Campylobacteria in the stomach and muscle material (% Campylobacteria: 34% and 32%, Fig. 1) supports a high reliance on carbon derived from rTCA. This specific separation contradicts the findings of Wang et al. (2014; 2022) who concluded that *X. testudinatus* mainly relies on photoautotrophic biomass. Our results support the hypothesis that *X. testudinatus* feeds on Campylobacteria to a greater extent than previously assumed.

## 4 Conclusions

Autotrophic carbon fixation at the Kueishantao shallow-water hydrothermal system, NE coast of Taiwan, is dominated by rTCA-utilizing microbes. In this study, we provide a first insight into the lipid-derived fatty acid contents and $\delta^{13}C$ values at the different vent environments. Fatty acids typically produced by sulfur-oxidizers (namely $C_{16:1\omega7c}$, $C_{18:1\omega7c}$, and $C_{18:1\omega9c}$) were accompanied by characteristically $^{13}C$-enriched isotope compositions originating mainly from rTCA utilizing chemoautotrophic Campylobacteria. Isotopic fractionation of biomass from rTCA utilizing microbes has previously been described to fall in the range of 2 to 13‰. So far, values close to 0‰ for fatty acids have only been reported once in isolated cultures (Elling et al. 2022). Here, we report ε values close to 0‰ for fatty acids in vent fluids with mixed origins, possibly extending the range of isotopic fractionation during rTCA. Likely, the driver causing low isotopic fractionation is the extremely low pH value, which is further supported by stronger $^{13}C$ enrichment in fatty acids at the YV site with lower pH values (pH 2.88) and higher fluid emission rates compared to samples from the less extreme WV site (pH 4.51). Additionally, higher trophic levels (*X. testudinatus*) also rely on the chemosynthetically fixed carbon, providing evidence of a vent-endemic interrelationship between chemoautotrophic Campylobacteria and higher organisms.

## 5 Appendix A: Experimental methods

One WV system at 10.5 m water depth and one YV system at 8 m water depth were sampled offshore Kueishantao in 2014 and 2015 (Fig. A1; Lin et al., 2019). The surface sediments at 1 m, 20 m, and 1200 m (background site M4) distances from the individual vent sites were retrieved in April 2014 during the OR2-2024 cruise on the RV Ocean Researcher II. The 20 cm long sediment cores directly from the individual venting sites, the vent fluids, the vent endemic crab (*Xenograpsus testudinatus*), and the filtered POC close to the vents (Mw and My series) and in the background area (B0-5) were retrieved during the OR2-2095 cruise (17-18 May 2015) on the RV Ocean Researcher II and one cruise that utilized fishing boats (25-28 May 2015; for more information see: Lin et al., 2019; 2020). The vent crab was collected within 30 m from the orifice of the white vent. Vent fluids were collected by inserting a polytetrafluoroethylene tube into the chimneys, connected via a

polytetrafluoroethylene valve to a pre-evacuated glass bottle. Sediment cores were taken using polycarbonate core liners (inner diameter 7 cm) with rubber end caps. They were transported back to the laboratory within a few hours after sampling in an upright position to ensure minimum disturbance of the sediment and subsampled in 2 cm depth intervals. For POC analyses, 5 to 14 liters of water were filtered from 6, 14, 20, and 40-meter away from the individual vents (WV: Mw series, YV: My series, background: B series, Fig. A1, Tab. A1). Each sample was collected using a 0.7 µm quartz filter. All samples were stored at -25°C until further analyses.

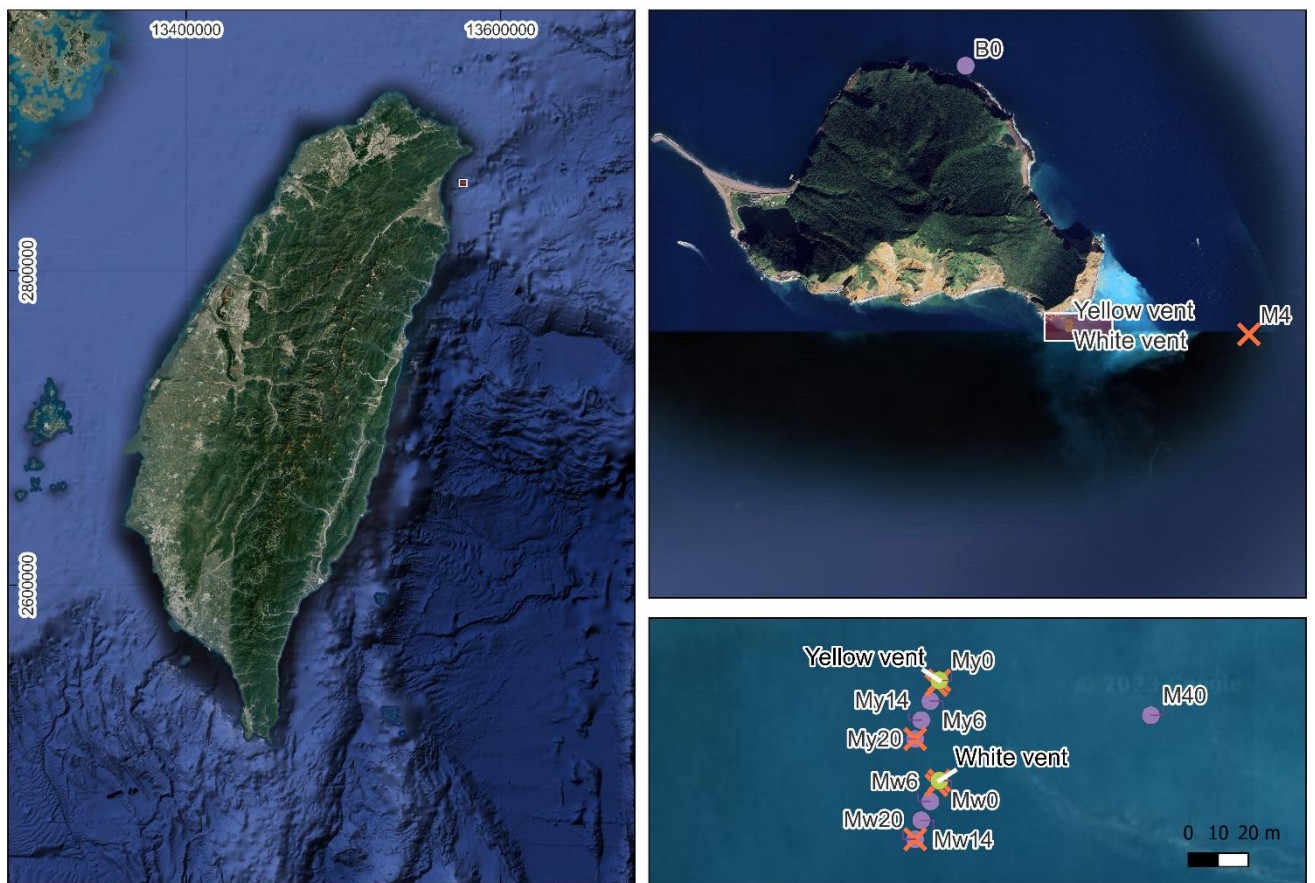

**Figure A1. Sampling location of the yellow vent (YV) and the white vent (WV) chosen for this study near Kueishantao off the NE coast of Taiwan (A, B). The map includes the location of the filtered particulate organic carbon (POC) from the water column (violet circles, My series influenced by the yellow vent, Mw series influenced by the white vent) and the background site (B0). Additionally, the sampling locations of the background sediment (M4) and the hydrothermally influenced sediments (YV core, YV 1 m, YV 20 m, WV core, WV 1 m, WV 20 m) are displayed in the map with orange crosses. Filter My20 corresponds to YV 20 m, filter Mw20 corresponds to WV 20 m. Sampling locations can be found in Lin et al. (2019). Map from © Google Satellite (2023).**

**Table A1. Overview of fluid and water samples collected near hydrothermal vent sites (YV = Yellow Vent, WV = White Vent) and at a background site. The table includes the sample series, name, distance to the vent, water depth, and filtered volume of each sample.**

| Vent | Series | Sample name | Distance to vent (m) | Water depth (m) | Filtered volume (L) |
|------|--------|-------------|----------------------|-----------------|---------------------|
| YV | My | My 0 (YV fluid) | 0 | 8 | 18 |
|    | My | My 6_5 | 6 | 5 | 4.9 |
|    | My | My 14_5 | 14 | 5 | 10.9 |
|    | My | My 20_5 | 20 | 5 | 5.3 |
| WV | Mw | Mw 0 (WV fluid) | 0 | 11.5 | 18 |
|    | Mw | Mw 6_5 | 6 | 5 | 12.3 |
|    | Mw | Mw 14_5 | 14 | 5 | 12.7 |
|    | Mw | Mw 20_5 | 20 | 5 | 13.8 |
|    | M (Transect) | M 40_0 | 40 | 0 | 11.2 |
| None | Background | B 0_5 | 1850 | 5 | 6.8 |

TO$^{13}$C (tissue material of *Xenograpsus testudinatus*) values of freeze-dried, decalcified, and homogenized samples have
been determined using a Thermo Scientific Flash 2000 Elemental Analyser connected with a Thermo Scientific Delta V Plus
IRMS via a ConFlo IV interface. Stable carbon isotope values were corrected using a laboratory $CO_2$ reference gas standard
and a certified international IAEA-CH-6 standard (e.g., Aepfler et al., 2022) and are reported in the δ-notation as δ$^{13}$C relative
to the Vienna Pee Dee Belemnite (VPDB) standard.

Samples for determination of fatty acid contents and their δ$^{13}$C values were extracted according to a modified Bligh and
Dyer protocol (Sturt et al., 2004). For later quantification of the fatty acids in the sediment cores and the vent fluids, an internal
standard (2Me-C$_{18:0}$ fatty acid) was added, all other samples (filtered POC, surface sediments, crab tissue) were quantified
using an injection standard containing *n*-C$_{18}$ to *n*-C$_{34}$ alkanes in known concentration. Saponification of an aliquot of the total
lipid extract was followed by the formation of fatty acid methyl esters (FAMEs) using boron trifluoride in methanol (Elvert et
al., 2003). The resulting FAMEs were analyzed using a Thermo Fisher Scientific Focus gas chromatograph equipped with a
30 m Restek Rxi-5ms column (0.25 mm internal diameter, 0.25 µm film thickness) and flame ionization detection (GC-FID)
for quantification. For identification, samples were either measured on a Thermo Finnigan Trace GC coupled to a Thermo
Finnigan Trace MS (GC-MS; vent fluids and sediment cores) or an Agilent 7820 GC coupled to a 5977E mass-selective
detector (GC-MSD; surface sediments and filtered POC), both equipped with the same column as for the GC-FID
measurements. Fatty acid $^{13}$C analyses were carried out using either a Thermo Scientific Trace GC coupled via a GC Isolink
interface to a Delta V Plus isotope ratio MS via a Conflo IV interface equipped with the same column as the GC-FID and the
GC-MS for vent fluids and sediment cores or a Thermo Scientific Trace GC equipped with a J&W DB-1 GC column (30 m
length, 0.25 mm i.d., 0.5 µm film thickness) coupled via a Conflo II to a Thermo Finnigan MAT 252 isotope mass spectrometer
for surface sediments and filtered POC.

Contents of fatty acids are given in µg g$^{-1}$ dried sediment or µg L$^{-1}$ filtered volume. We calculated the proportion of
Campylobacteria from fatty acids known to be produced by sulfur oxidizers (e.g., Zhang et al., 2005; Wang et al., 2021):
C$_{16:1\omega7c}$, C$_{16:1\omega7t}$, *n*-C$_{16:0}$, C$_{18:1\omega9c}$, C$_{18:1\omega7c}$, and C$_{18:1\omega7t}$. The δ$^{13}$C values of these fatty acids were used to attribute them to
Campylobacteria, with -7.2‰ from the vent fluids as an endmember for rTCA and -27‰ from the mean of sediment

background fatty acids ($C_{16:1\omega7c}$, $C_{18:1\omega9c}$, $C_{18:1\omega7c}$) as the endmember for CBB-utilizing microbes. Based on these proportions, we determined the overall percentage of Campylobacteria-related fatty acids as a percentage of the total concentration of all fatty acids detected. All $\delta^{13}C$ values are expressed relative to VPDB, have been determined relative to the laboratory $CO_2$ reference gas with an analytical error of <0.3‰, and were corrected for the addition of carbon from methanol (MeOH) during derivatization using the following equation:

$$\delta^{13}C_{FA} = \frac{(\delta^{13}C_{FAME} * (n_{FA} + n_{MeOH})) - (\delta^{13}C_{MeOH} * n_{MeOH})}{n_{FA}}$$

To distinguish samples with different degrees of hydrothermal influence and percentage of rTCA carbon uptake, a cluster analysis using the Bray-Curtis similarity matrix was performed on the $\delta^{13}C$ values of most abundant and diagnostic fatty acids (i.e., $C_{16:1\omega7c}$, $n$-$C_{16:0}$, $C_{18:1\omega7c}$) in all samples. The type of cluster analysis used was UPGMA and no transformation was performed on the isotope data. A similarity profile test (SIMPROF) was applied to identify significant differences between the clusters (Clarke et al., 2008). For the SIMPROF test, 1000 permutations were applied and the chosen significance level was 0.05. The cluster analysis was performed in MATLAB R2021b using the Fathom toolbox (Jones, 2017).

## Data availability

The supplementary data supporting the findings of this study is available in the PANGAEA database https://doi.pangaea.de/10.1594/PANGAEA.967575 (Maak et al., 2024, a moratorium is in place until this manuscript is published).

## Author contribution

Samples were collected by YSL, SIB, LLL, and RFA. JMM, SIB, ME, ES, and RFA designed the experiment, and JMM and RFA carried it out. JMM interpreted the data and prepared the manuscript with contributions from all co-authors.

## Competing interests

The authors declare that they have no conflict of interest.

## Acknowledgments

We thank the Taiwan Ministry of Environment for the access permission to the samples. Further, we thank the scientists, crews, and scientific divers of the OR2-2024 and OR2–2095 cruises on the RV Ocean Researcher II and one cruise that utilized fishing boats (25–28 May 2015) for sampling. This research was funded by Deutsche Forschungsgemeinschaft (DFG, German Research Foundation) under Germany's Excellence Strategy—EXC-2077—390741603, a DFG Emmy Noether Grant donated

to SIB (BU 2606/1-1), and the DAAD (PPP program for project-related personal exchange - 57138084, given to SIB and YSL). We acknowledge financial support from the Open Access Publication Fund of the University of Bremen.

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
