# Peer review of "The energy-efficient reductive tricarboxylic acid cycle drives carbon uptake and transfer to higher trophic levels within the Kueishantao shallow-water hydrothermal system"

_EGUsphere, 2024_

## Author Response (AR1)

Reviewers comments:

**Reviewer #1:**

This reviewer cannot fault the paper; it's well written and the analytical work appears to be very solid. The only suggestion I offer before the work is published is to include, in the methods section, an explicit statement as to how the FAME values were corrected back to FA for the carbon added during derivatization.

Response: We sincerely thank you for your positive evaluation of our manuscript and your kind words about the quality of the writing and analytical work. In the revised manuscript, we have included an explicit statement including an equation in the Methods section explaining how the $\delta^{13}C$ values of fatty acid methyl esters (FAMEs) were corrected back to fatty acids (FA, line 240 ff.).

**Reviewer #2:**

The authors conducted a study on the stable carbon isotope composition of fatty acids from samples collected around shallow-water hydrothermal vents near Kueishantao, Taiwan. Certain fatty acids (C16:1ω7c, C18:1ω9, C18:1ω7c) were notably 13C-enriched and, thus, closer to the measured d13C-DIC values in the water column than other compounds. These fatty acids are known to be produced by sulfur-oxidizing Campylobacteria that are chemoautotrophic microbes using the reductive tricarboxylic acid (rTCA) pathway, which tends to be associated with considerably lower isotopic fractionation during carbon fixation than the Calvin-Benson-Bassham cycle. Consequently, the authors pointed out, the 13C-enriched fatty acids are most likely derived from microbes utilizing the rTCA. The authors further argue that this study provides the first field evidence of lipid synthesis with negligible isotopic fractionation. Their sampling approach, which included sediments, vent fluid, water column particulate organic carbon, and a crab specimen, allowed them to demonstrate a decline in both the abundance and 13C enrichment of these fatty acids with increasing distance from the vent systems.

Overall, this is a well-executed and informative study. I found these shallow-water hydrothermal vents fascinating and appreciated learning about them through this work. I believe that the manuscript could benefit from additional information about the use and limitations of the mass balance model and how potential variations in isotopic fractionation during biosynthesis of different compounds may affect their interpretation. I hope the authors will find the comments below helpful.

Response: We sincerely thank you for your thorough review and insightful feedback on our manuscript. We greatly appreciate your time and effort in evaluating our work and providing constructive suggestions. Below, we address each of your points in detail.

-- Is it correct that lipids tend to go through greater isotopic fractionation than other biomolecules, such as proteins and carbohydrates, during biosynthesis? If so, how appropriate it is to draw "hard lines" for rTCA and CBB range of fractionation on Figure 3 and assign specific fatty acids to the closest carbon fixation pathway? Perhaps elaborating on the

limitations and uncertainties would be beneficial. Also, would any differences in the extent of isotopic fractionation introduce a level of uncertainty into the mass balance model?

Response: Good point. In the revised manuscript, we will elaborate in the Discussion that lipids undergo greater isotope fractionation compared to other biomolecules (e.g., DeNiro and Epstein 1977, https://doi.org/10.1126/science.327543). To address your concern about the "hard lines" in Figure 2, we have revised the figure caption to explicitly acknowledge the uncertainties in assigning fatty acids to specific pathways by adding "Since lipids can exhibit greater isotope fractionation than other biomolecules such as proteins and carbohydrates (DeNiro and Epstein, 1977), a larger isotope fractionation than the indicated ranges is possible." (line 109 f.)

Additionally, we have clarified in the Discussion that while the isotope ranges provide a valuable framework, natural variations in isotope fractionation may introduce uncertainties in using "hard lines" by adding "In theory, lipids can exhibit greater isotope fractionation than proteins and carbohydrates due to further biosynthetic processes (DeNiro and Epstein, 1977). Despite this potential variability, the indicated ranges provide a valuable framework for estimating whether samples or specific fatty acids can be assigned to particular carbon fixation pathways. " (line 127 ff.)

-- Are bulk stable carbon isotopic values available for any of the samples, especially for the crab specimen? It seems like that previous work about this endemic crab used bulk isotopes. I think it would be helpful to report the bulk values (if available) for this crab specimen to make this work more complementary and comparable to previous studies. Especially because the authors propose a different primary food source based on their findings than previous studies (lines 158-159). Here, acknowledging that this study used a single specimen, and consider the likely variability in this omnivorous(?) crab species would be appropriate.

Response: Yes, we do have bulk stable carbon isotopic values for the crab specimen and have added the resulting bulk $^{13}$C values in the results section (line 66 f.), as well as discussed the resulting bulk isotope values (line 163 ff.). The Experimental Methods were also extended to include the information about the TO$^{13}$C measurements (line 214 ff.).

-- The description and interpretation of the mass balance model requires some clarification. This mass balance was used to provide an estimate for the relative abundance of Campylobacteria using lipid biomarkers and their d13C. First, I think it would be fair to the reader to give more information in the Figure 1 caption than just "% Campylobacteria in fatty acids", I found it confusing. Would calling the axis label "potential Campylobacteria % based on FA proxies" on Fig. 1 plot-D and -G be more appropriate? Moreover, the mass balance is poorly described in the methods. Does it account for isotope values only? Or the relative abundance of the selected lipids is factored into the calculations?

Response: To clarify, the calculations included the concentration of all detected fatty acids, but only specific fatty acids (C$_{16:1\omega7c}$, C$_{16:1\omega7t}$, n-C$_{16:0}$, C$_{18:1\omega9c}$, C$_{18:1\omega7c}$, and C$_{18:1\omega7t}$) were assigned to Campylobacteria. The first step involved calculating the percentage of these fatty acids attributable to Campylobacteria based on their $\delta^{13}$C values. Afterward, we calculated the overall percentage of Campylobacteria-related fatty acids as a proportion of the total concentration of all fatty acids detected. We have revised the Methods section to provide a more detailed explanation of this calculation and ensure that readers fully understand the

approach (line 234 ff.). We changed the axis label in Figure 1 to "% Campylobacteria based on FA proxies".

My understanding is that only those fatty acids were selected for the mass balance that are known to be synthesized by sulfur-oxidizer Campylobacteria. Wouldn't such an approach inherently introduce a bias toward Campylobacteria? If yes, wouldn't such statements like the one starting on line 156 would qualify as a circular argument? Line 156-158: "Further, a high lipid content originating from Campylobacteria in the stomach and muscle material (% Campylobacteria: 34% and 32%, Fig. 1) supports a high reliance on carbon derived from rTCA."

Response: The statement starting on line 156 ("Further, a high lipid content originating from Campylobacteria...", now line 171) is not circular because the calculation process is based on a quantitative mass balance that incorporates both $\delta^{13}C$ values and the total fatty acid concentrations, rather than selectively considering only those fatty acids synthesized by Campylobacteria. This methodology avoids bias toward Campylobacteria and ensures that the conclusion is supported by the data.

-- It took me a while to get an overview of the layout of the sampling stations and understand the sampling design. Personally, I would appreciate more clarity and more visibility about the spatial distribution of the collected samples because it is an important aspect of the study. This issue could be solved by placing a plot-B in the appendix Figure A1 showing the spatial distribution of the collected samples along the transect, including sediment, the crab, and POC. As of now, the reference to see Lin et al for more info is difficult to follow and that study had more stations/sites, so it is not easy to tease out what is applicable here.

Response: Thank you for this suggestion. We have included a new plot in the appendix showing the spatial distribution of the sampling sites (line 204 ff.), with clear labeling of the sediment and POC collection points. As noted in the preprint, the crab was collected within 30 m of the white vent orifice; unfortunately, more precise coordinates are not available.

-- Perhaps a plot about the sampling layout would also help to place and keep track of all the site acronyms M4, B0...etc. The "My and Mw" were initially called water column samples (Figure 1, 2) than the code names were introduced later (Figure 3) and so on. This approach makes it hard to follow for someone who is not familiar with the spatial distribution of the stations/sites (like me).

Response: We have added a table in the Appendix to enhance the understanding of the reader which sample was taken at which distance to the venting sites, including the vent, series, sample name, distance to vent (m), water depth (m), and filtered volume (L) (line 211 ff.)

-- POC: how many samples and how many liters were filtered?

Response: we have included a brief paragraph in the Experimental Methods section (starting line 200 ff.) and an overview in the form of a table (line 211 ff.) to make sure the reader understands how many samples were used and how many liters were filtered.

-- I was curious to check the supporting data to get a better overview of the collected samples, and in the hopes that I may learn something about the mass balance, but it is under moratorium. Although the preprint site says reviewers have access, I couldn't access it. Also, it seems like it is available through a members-only open access platform that requires personal data to log in (as long as it is in line with BG policy, I have no objection).

Response: We already contacted the editor of the journal.

-- Given the authors argue it is the first field observation of negligible isotopic fractionation during rTCA, it would be more convincing if they elaborated on why these environmental conditions might result in the observed isotopic effects. Such low pH will certainly shift the DIC dynamics toward CO2 dominance...

Response: The reviewer is correct, due to the low pH in the system, $CO_2$ (dissolved) dominates the DIC pool. Overall, the dissolved inorganic carbon (DIC) concentrations within the vent fluids are very high: 4340 µmol/kg at the white vent and 2450 µmol/kg at the yellow vent. In comparison, samples taken approximately 80 meters from the vent (M80) show concentrations of 2300 µmol/kg (all DIC concentrations from Lin et al., 2019; https://doi.org/10.1016/j.marchem.2019.02.002).

While the system experiences significant $CO_2$ degassing, the elevated DIC concentrations are likely sustained by a constant supply of carbon-rich fluids from below. We will expand on this explanation in the manuscript by mentioning "DIC concentrations at the hydrothermal vents are in excess (Lin et al., 2020), despite the acidic pH values that typically promote $CO_2$ degassing. The elevated concentrations are likely sustained by a constant supply of carbon-rich fluids from below." (line 133 ff.)

To our knowledge, how and why low pH values or other environmental parameters, such as temperature, affect isotope fractionation has not yet been investigated.

Minor comments:

-- line 127: "therefore" should be "we argue" or similar, and the following line has T mentioned twice. Why not call it temperature for clarity?

Response: Thanks for pointing that out; we have changed it to "We argue" (line 138). T was changed to temperature (line 139 f.).

-- Figure 2 is impossible to interpret in its entirety. Pooling similar sites/samples and show them with the same symbol or using other ways to simplify the figure would help clarity. The point is made that certain FAs exhibit notably 13C-enriched values, but teasing out where they come from is hard to see and interpret.

Response: We have simplified Figure 2 by merging the sediment data from the individual venting sites and consolidating the data for *Xenograpsus testudinatus*. This adjustment reduced the total number of displayed samples by five (line 104).

-- line 142: there is no Figure 4

Response: Changed to Fig. 3 (line 153)

Response: Thanks for noticing, we have changed it to trophic transfer (line 168)

**Note**:

We have changed $C_{18:1\omega9}$ in $C_{18:1\omega9c}$ to clarify the double bond position and did minor changes on the text to enhance clarity (line 18 "enrichment" was changed to "signal")